

# Snow data assimilation for seasonal streamflow supply prediction in mountainous basins

Sammy Metref[1], Emmanuel Cosme[1], Matthieu Le Lay[2], and Joël Gailhard[2]

[1]Université Grenoble Alpes, Centre National de la Recherche Scientifique, Institut de Recherche pour le Développement, Institut des Géosciences de l'Environnement, Grenoble, France.
[2]Électricité de France – Division Technique Générale, Saint-Martin-le-Vinoux, France.

**Correspondence:** Sammy Metref (sammy.metref@univ-grenoble-alpes.fr)

**Abstract.** Accurately predicting the seasonal inflow into a reservoir accumulated during the snowmelt season, for instance the total aggregated inflow between April and August (A48), in a hydrological basin is critical to anticipate the operation of hydroelectric damns and avoid hydrology-related hazard. Such forecasts generally involve numerical models that simulate the hydrological evoluation of a basin. The operational department of the French electric company EDF implements a semi-
distributed model and carry out such forecasts for several decades, on about fifty basins. However, both scarse observation data and over-simplified physics representatioin may leed to significant forecasts errors. Data assimilation has been shown beneficial to improve predictions in various hydrological applications, yet very few have addressed the seasonal streamflow supply prediction problem. More specifically, the assimilation of snow observations, though available in various forms, has been rarely studied, despite the possible sensitivity of the streamflow supply to snow stock. This is the goal of the present
paper. In three mountainous basins, a serie of four ensemble data assimilation experiments – assimilating (i) the streamflow (Q) alone, (ii) Q and fractional snow cover (FSC) data, (iii) Q and local cosmic ray snow sensor data (CRS) and (iv) all the data combined – are compared to the climatologic ensemble and an ensemble of free simulations. The experiments compare the accuracy of the estimated streamflows during the reanalysis (or assimilation) period, September to March; during the forecast period, April to August; and the A48 estimation. The results show that Q assimilation notably improves streamflow estimations
during both reanalysis and forecast period. Also, the additional combination of CRS and FSC data to the assimilation further ameliorates the A48 prediction in two of the three basins. In the last basin, the experiments highlight a poor representativity of the CRS observations during some years and reveals the need for an enhanced observation system.

**Keywords.** Mordor-SD, Hydrological runoff, CRS, SWE, snow cover, MODIS, particle filter, long-term forecast.

## 1 Introduction

The seasonal streamflow supply (A48) forecast, i.e. predicting the total amount of water at the basin outlet during the snowmelt period between April and August, is a crucial information needed for instance to operate and manage hydroelectric damns. Hence, the operational department of the French electric company EDF carry out such forecasts for several decades, on about fifty basins. Yet, in mountainous basins, the confidence provided by long term hydrological forecast is affected by the uncer-



tainty on the meteorological forcings (Li et al., 2009; Bormann et al., 2013; Luce et al., 2014) and the inaccurately simulated

snowpack (Liston and Sturm, 1998; Pan et al., 2003). Acknowledging that the seasonal streamflow supply partly depends on the snowpack accumulated during winter, the growing number of satellite observations of snow-related quantities and *in situ* snow measurements may open the way to improving the A48 predictions in mountainous basins. The relevance of using such observations is an open question though: (i) How much is the A48 prediction sensitive to the snowpack? (ii) Do the snow observations contain enough information to estimate the snowpack accurately enough to impact the quality of predictions?

And (iii) How do existing analysis methods perform in estimating the snowpack from the snow observations?

Some studies suggest that controlling the snowpack evolution using observations can significantly ameliorate short and long term streamflow forecast (Viviroli et al., 2011; Fayad et al., 2017). In the present paper, a sensitivity experiment is conducted to highlight how the uncertainties propagate within a hydrological system. The experiment calculates the Sobol indices for each of the model variables, indicating the impact that the uncertainty of these variables has on the uncertainty of the streamflow

at the outlet. The experiment confirms that snow stocks play an important role on streamflow uncertainty. Hence, a better representation of the snowpack should result in a significant gain in seasonal streamflow supply estimation.

Data assimilation techniques are often used to help control and refine hydrological systems (see Largeron et al., 2020 for a detailed review). Several studies have successfully assimilated snow water equivalent (SWE) data but mostly in local models, i.e., models describing the snow dynamic at a specific site and not the hydrological system of an entire basin. Indeed,

SWE measurements, especially from ground-based cosmic ray sensor (CRS; Kodama et al., 1979; Paquet and Laval, 2006) instruments, provide very local information which can be used to improve a local model at a specific site (e.g., Piazzi et al., 2018 in three Alpine sites). However, assimilating CRS data in a basin scale model as is can lead to representativity errors (where the SWE measured by CRS does not correspond to any relevant global SWE model), thus deteriorating the system estimation. A test (not shown here) has been performed to assess a CRS assimilation directly in the hydrological system and

did indeed strongly deteriorate the streamflow estimation. To circumvent this issue an alternative approach to consider CRS data in a basin scale model is discussed in Section 4.3, used throughout the following experiments and shows promising results.

Multiple studies have implemented ensemble-based data assimilation schemes, such as the ensemble Kalman filter (EnKF, Evensen, 2003), of direct or indirect snow observations (Andreadis and Lettenmaier, 2006; Clark et al., 2006; Slater and Clark, 2006; Su et al., 2008; Magnusson et al., 2014; Piazzi et al., 2019, 2021). However, the nonlinear nature of these snow related

observations as well as the complexity to control a hydrological system with indirect information seem to favor the use of a more nonlinear and non-Gaussian data assimilation method, especially when aiming at long lead time prediction improvements (Dumedah and Coulibaly, 2013). One data assimilation method in particular, the particle filter (PF, Van Leeuwen, 2009), is known for its ability to handle highly non-linear systems containing non-Gaussian probabilities. Indeed, the PF applies Bayes' theorem by describing the probability density functions as a sum of Dirac from an ensemble of simulations (particles)

and without any additional hypothesis. Therefore, under the assumption of a sufficiently large ensemble of particles, the PF provides the optimal solution of any inverse problem. In hydrological applications, DeChant and Moradkhani (2011) managed to improve SWE and discharge forecast using microwave radiance assimilation with a PF. Also, Leisenring and Moradkhani (2011) showed in a synthetic experiment comparing an EnKF and a PF, that the assimilation of SWE data with a PF improved





seasonal predictions. The work of Charrois et al. (2016) has shown the good performance of the PF for the assimilation of
optical reflectivity and snow depths and Piazzi et al. (2018) successfully used a PF for SWE data assimilation in moutainous
regions. Finally, Piazzi et al. (2021) concluded that PF assimilation outperforms an EnKF assimilation by generating longer-
lasting predictions.

The objective of the present paper is to assess the potential of using local snow observations in a seasonal forecast procedure
to improve the streamflow supply prediction at the outlet of mountain basins. This is addressed by implementing real data
assimilation experiments.

These experiments are based on the MORDOR-SD model (Garavaglia et al., 2017), the semi-distributed version of the origi-
nal MORDOR model, used by EDF for many years. The experiments have been deployed on three French mountainous basins.
Three types of observations are available in these basins: the observed streamflow at the outlet Q, cosmic ray snow sensor CRS
data and fractional snow cover (FSC, Masson et al., 2018), provided by the moderate resolution imaging spectroradiometer
(MODIS) satellite. Each year, an assimilation of the available data is performed from September to March of the following
year. Throughout the paper, this time period is called the reanalysis (or assimilation) period. A free forecast is then run from
April to August. This time period is called the forecast period. The performance of the assimilation is evaluated during both
the reanalysis and the forecast period.

The results indicate that data assimilation significantly improves the seasonal streamflow supply prediction, especially when
the observations are combined, except in one basin during specific years. The results are evaluated using an ensemble-based
score called the continuous rank probability score skill (see Hersbach, 2000 for details on the CRPS and Piazzi et al., 2018
for details on the CRPSS) and a deterministic score: the root-mean-square errors (RMSE). The scores obtained show that the
streamflow Q data assimilation systematically improves the streamflow reanalysis, i.e. the period from September to March
when observations are available ; the streamflow prediction during the forecast period, from April to August and the A48
estimation. Moreover in the Verdon and Naguilhes basins, the combined assimilation of Q, CRS and FSC data further refines
the A48 estimation, by providing a better initialization of the snow stocks for the forecast at the end of March. In the Guil
basin, the free ensemble A48 prediction is already accurate and the assimilation of streamflow and FSC does not improve it
much. However, the CRS assimilation not only does not improve but deteriorates the A48 prediction during specific years.
This result highlights an inconsistency between the streamflow observations and the CRS observations during those years. This
information can be a crucial indication that a more representative observation system is required in that hydrological basin.

The paper is divided into five parts: a description of the hydrological system, i.e. the numerical model, the three hydrological
basins and the available observations (Section 2); a study of the sensitivity of the system (Section 3); the description of the
experimental protocol (Section 4) and the assimilation results (Section 5). A summary and conclusions are drawn in Section 6.



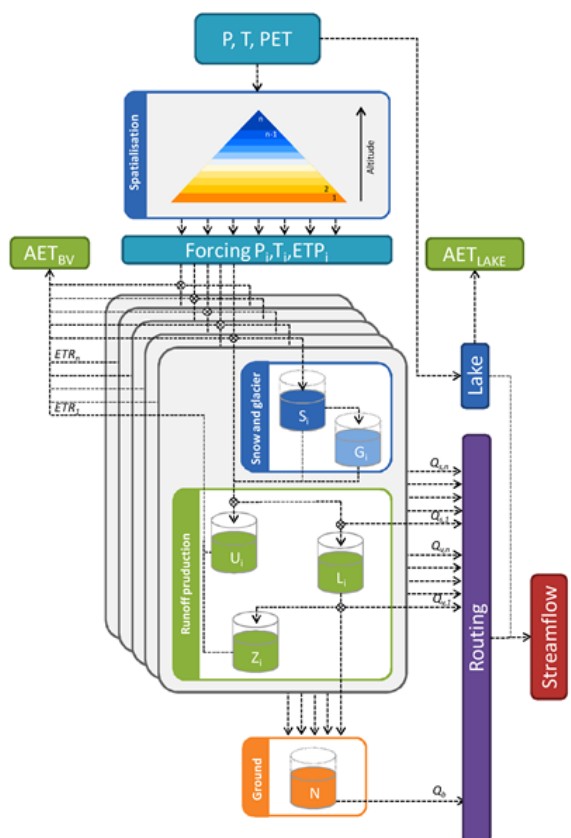

**Figure 1.** [From Garavaglia et al. (2017)] General scheme describing the MORDOR-SD model.

## 2 Hydrological system

### 2.1 Mordor-SD model

For many years, EDF teams have been using a hydrological box model: the Mordor model. The semi-distributed MORDOR-SD model (Garavaglia et al., 2017) is an improvement on the original MORDOR that includes a spatial discretization scheme. This discretization is based on an elevation band approach, adapted for mountain hydrology. The number of elevation bands depends on the studied mountainous basin. In most MORDOR-SD applications, the spatial variability of meteorological forcing is summarized by two orographic gradients, one for precipitation and the other for temperature. In this way, we assume that in mountainous areas, spatial variability is primarily determined by elevation. Most of the state variables in the model are calculated for each elevation band. Only groundwater content and streamflow runoff are considered global and are calculated at the basin scale. A diagram from Garavaglia et al. (2017) describing the components and flows of MORDOR-SD is shown





**Figure 2.** Time series of the forcings: precipitation P (top) et the temperature T (bottom) during the year 2001-2002 in the Verdon basin. The deterministic forcings are represented in blue and the corresponding perturbed ensemble is represented in gray.



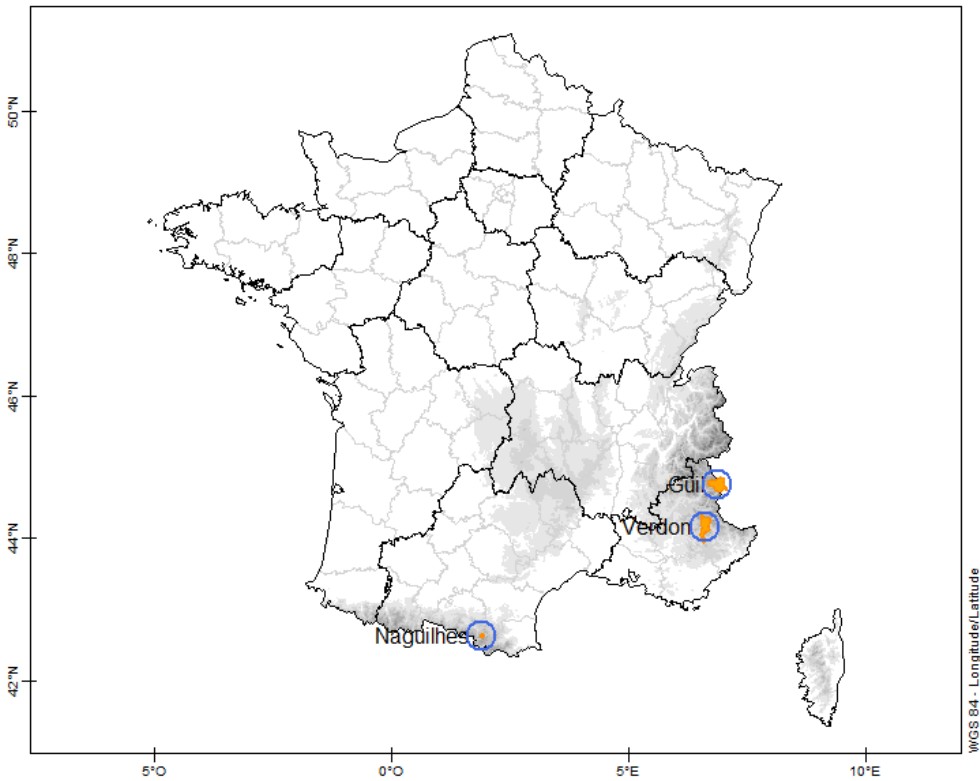

**Figure 3.** Geographic locations of the Verdon basin, the Naguilhes basin and the Guil basin.

in Figure 1. In this study, Mordor-SD configuration has 10 to 12 free parameters, 5 state variables in each elevation band and 2

global variables.

In addition to the state variables, the Mordor-SD model depends on two atmospheric forcings: temperature T and precipitation P. Both forcings are computed from local meteorological measurements and prescribed at daily time steps. As discussed previously, the model modifies the impact of the forcings at the different elevations using two orographic gradients. In the rest of the work presented here, these gradients are constant and will not be discussed further. However, these vertical gradients

might represent a significant source of uncertainty and their impact should be investigated in future works.

Each year, the model has one deterministic time serie for temperature and one for precipitation. In order to better represent the uncertainty in these forcings, an implementation of first-order stochastic auto-regressive processes (AR1) was performed in order to introduce consistent perturbations in time. An AR1 process is added to the temperature in order to simulate the instrument and the representativity errors on the forcing. Whereas, the AR1 process for the precipitation is centered around 1

and then mutliplied, thus the variability in the precipitation intensity is simulated but no new days of precipitation are created. An exemple of the ensemble of forcings generated for the year 2001 in the Verdon basin is illustrated in Figure 2.





These stochastic processes allow to generate ensembles of simulations which describe possible realizations of the atmospheric state. The calibration of these ensembles (i.e., calibration of the parameters of the auto-regressive processes) play a crucial role in the implementation of the assimilation system and is further discussed in Section 4.1.

**Figure 4.** Available observation time series in the Verdon basin (left), in the Naguilhes basin (center) and in the Guil basin (right) of streamflow Q (top), CRS SWE observation (center) and FSC (bottom).

## 2.2 Hydrological basins and observations

The present study focuses on three mountainous basins: the Verdon at La Mure basin, the Naguilhes basin and the Guil at Chapelue basin (Figure 3). These basins are part of the EDF hydroelectricity network.

The Verdon at La Mure basin is a sub-basin of the Durance basin located in the Southern French Alps. The Verdon basin covers 404 $km^2$ and has an elevation ranging from 972m to 2990m. The Naguilhes basin is located on a tributary of the Ariege river in the Eastern part of the French Pyrenees. It is the smallest of the studied basins, covering 30 $km^2$ and with an elevation





ranging from 1880 to 2750m. The basin corresponds to the inflow from the Naguilhes hydroelectric damn. The Guil basin is a tributary of the Durance river, located in the French Alps (*Hautes-Alpes*). The Guil at Chapelue basin covers 418 $km^2$ and has an elevation ranging from 1313m to 3274m. The outlet is located just upstream from Maison du Roy damn.

In each basin, the streamflow data were collected by EDF. The precipitation and temperature are computed from a statistical
reanalysis based on ground network data and weather patterns (Gottardi et al., 2012). These data are then used to calibrate and force the model.

In this study, three types of observations are available in the basins: the streamflow, the CRS and the FSC.

The streamflow is the observed water flow at the basin outlet given in $m \cdot s^{-1}$. The streamflow is a direct and reliable observation of the model state variable Q. The streamflow data are available almost continuously since 1997 in the Verdon
basin, 1962 in the Naguilhes basin and 2004 in the Guil basin.

The CRS (Kodama et al., 1979; Paquet and Laval, 2006) is a cosmic ray snow sensor located in every basin as part of the EDF snow network, and provides the *snow water equivalent* (SWE) that informs on the state of the snow stock at a specific geographical point.

- In the Verdon basin, the instrument is located at the Sanguignères station (V2804) at an altitude of 2050m. The CRS data
are available discontinuously from 2002 to 2017.

- In the Naguilhes basin, the instrument is located at the Les Songes station (V4322) at an altitude of 2030m. The CRS data are available discontinuously from 2004 to 2017.

- In the Guil basin, the instrument is located at the Les Marrous station (V2471) at an altitude of 2730m. The CRS data are available discontinuously from 2005 to 2016.

The CRS provide a very local information and might not always be representative of the snow evolution on the entire basin. In Section 4.3, a detailed discussion is held on how the CRS observations are integrated in the assimilation process.

The FSC is provided by the MODIS satellite observations (Hall et al., 2006). The FSC is quantified by a value ranging from 0 to 1, for zero to full coverage. The FSC data are available discontinuously (depending on cloud cover) from 2001 to 2015 in the Verdon, from 2003 to 2015 in the Naguilhes and from 2002 to 2015 in the Guil basin.
The three types of observations are displayed for each basin in Figure 4.

## 3 Sensitivity experiment

### 3.1 Sobol indices

In order to better understand the sensitivity, and thus the controlability, of the Mordor-SD model, we seek to determine which variables generate the most uncertainty in the streamflow estimate at the basin outlet. To do so, we perform a sensitivity study
of the system based on the Sobol indices (Sobol', 1990; Nossent et al., 2011).





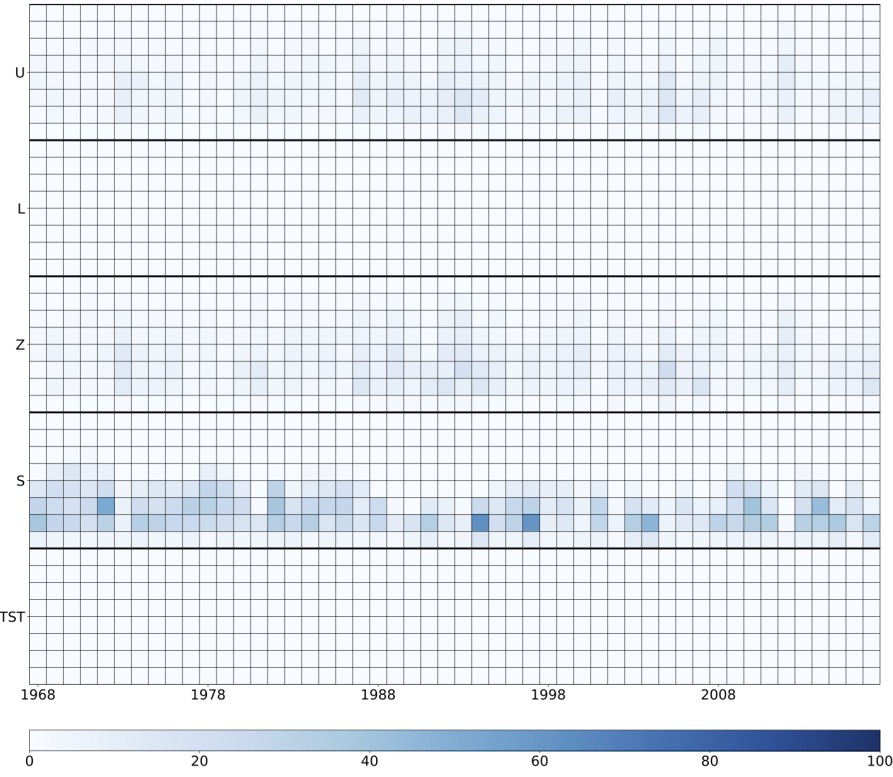

**Figure 5.** Sobol indices (in %) for a 10% perturbation in Verdon.

The Sobol indices evaluate the sensitivity of an output variable to an input variable. If a model links one or more random variables $X_i$, $i \in [1, n]$ (input variables) to one random variable Y (output variable), the Sobol index (of first order) of the variable $X_i$ is based on a variance decomposition and is defined by :

$$S_i = \frac{\mathbf{Var}\left[\mathbf{E}\left[Y|X_i\right]\right]}{\mathbf{Var}\left[Y\right]}. \tag{1}$$

**3.2 Mordor-SD sensitivity**

In the case of the Mordor model, one can see the A48 value as an output variable and all other state variables of the model as input variables. It is then possible to run a set of ensemble simulations by perturbing each variable independently to compute $\mathbf{Var}\left[\mathbf{E}\left[Y|X_i\right]\right]$ and another set by perturbing all the variables at once to compute $\mathbf{Var}\left[Y\right]$. This gives the A48 sensitivity to each state variable in the model.

To carry out this sensitivity study, a set of simulations is generated in each basin on April 1st of each year and the impact on the seasonal streamflow supply on August 31st is evaluated. Figure 5, 6 and 7 show the Sobol indices (in percent), in the



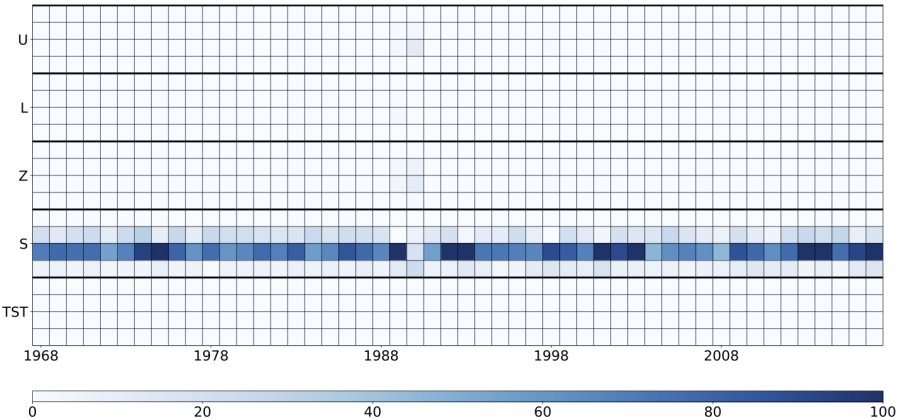

**Figure 6.** Sobol indices (in %) for a 10% perturbation in Naguilhes.

Verdon, the Naguilhes and the Guil basins respectively, for a perturbation on each variable of 10% of its value. It can be seen that for all years the variable uncertainties that lead to the largest uncertainties in cumulative streamflow are the uncertainties on snow stocks at the altitude bands from $S_4$ to $S_7$ in the Verdon, from $S_2$ to $S_4$ in the Naguilhes and from $S_4$ to $S_7$ in the Guil

basin. This result confirms that the most important lever to improve the streamflow estimate is the control of the snow stocks.

## 4 Experimental protocol

### 4.1 Protocol and diagnostics

The experiments are performed during the years when CRS and streamflow observations are available: from 2002 to 2017 in the Verdon basin, from 2004 to 2017 in the Naguilhes basin and from 2005 to 2016 in the Guil basin.

Every year, data assimilation is performed between September, 1st and March, 31st, this period is called the reanalysis period. The assimilated ensemble is then forecasted freely from April, 1st to August, 31st, this period is called the forecast period. The streamflow estimations are diagnosed during both the reanalysis and the forecast period. The A48 estimation, i.e., the cumulated streamflow during the forecast period, is also diagnosed.

The diagnostics performed are the *continuous rank probability score skill* (CRPSS; see Hersbach, 2000 for details on the

CRPS and Piazzi et al., 2018 for details on the CRPSS) according to the formulation described by Bontron (2004), with a thinness component (FinS) and a correctness component (JustS). A score of 1 represents a perfect ensemble and lower than 0 an ensemble less accurate than the climatology of the system. The FinS score can be seen as a measurement of the dispersion of the ensemble and the JustS a distance between the median of the ensemble and the observations. A second diagnostic is used





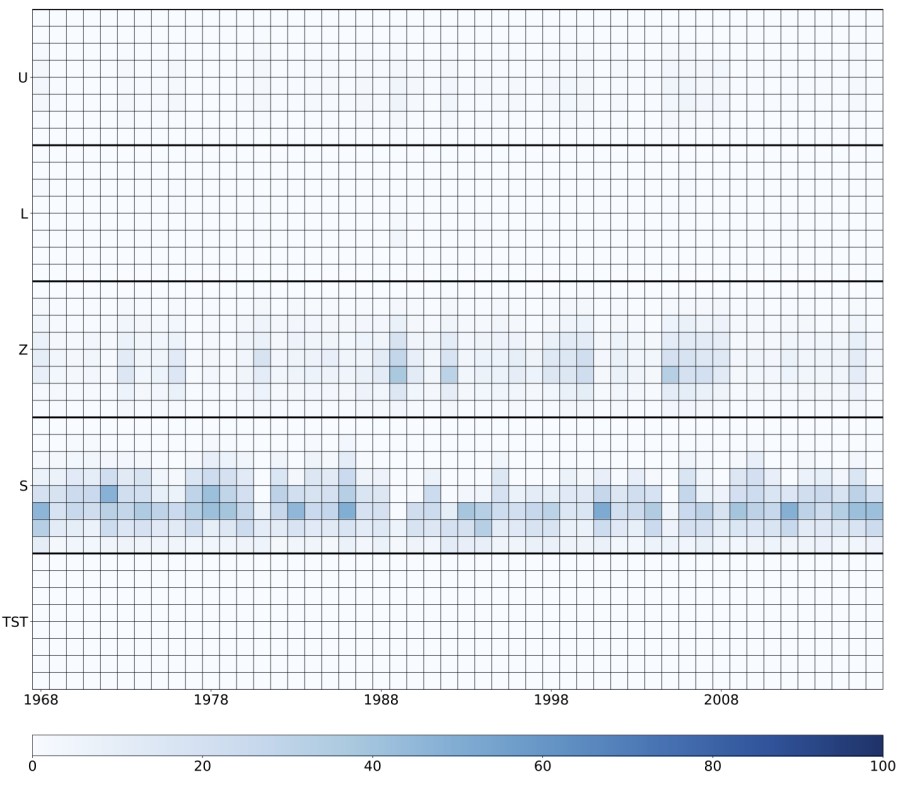

**Figure 7.** Sobol indices (in %) for a 10% perturbation in Guil.

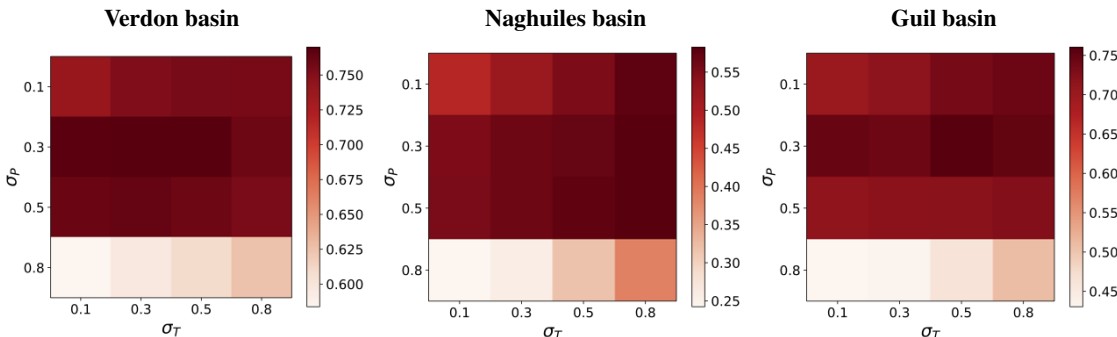

**Figure 8.** CRPSS of free ensemble simulations computed for AR1 parameter calibration $(\sigma_P, \sigma_T)$. The maximum CRPSS occurs at Verdon basin for (0.3,0.3), at Naguilhes basin for (0.3,0.8) and at Guil basin for (0.3,0.5).



to assess the A48 estimation: the root-mean-square error (RMSE). The RMSE is the euclidian distance between the ensemble
mean A48 estimation and the observed A48 and is computed, here, in $hm^3$. A perfect RMSE score is equal to 0.

## 4.2 Meteorological forcing perturbations

The free ensemble and the assimilation ensemble are composed of 900 members (simulations) that were generated using AR1
processes on the forcings. The AR1 autocorrelation parameters are prescribed for all experiments as 0.9 for temperature and
0 for precipitation. Note that the AR1 process applied on precipitation is multiplicative and the one applied on temperature is
additive. The AR1 standard deviations for the free ensemble were tuned to provide the most accurate A48 prediction. Figure 8
shows the CRPSS on the A48 estimation for free ensembles with several sets of AR1 standard deviations parameters $(\sigma_P, \sigma_T)$
applied to the forings (P,T).

A reproducibility issue was encountered during the assimilation experiments (several experiments with the same parameters
produced different results) probably due to the high non-linearities of the system and the finite number of ensemble members.
To avoid this problem, the standard deviations $\sigma_P$ and $\sigma_T$ of the AR1 processes on the forcings used for the assimilation were
increased to stabilize the results, during the reanalysis period. Then, during the forecast period, the assimilation ensemble uses
the same AR1 process parameters as the free ensemble. Table 1 summarizes the AR1 parameters used in the experiments.

|  | Verdon basin | | Naguilhes basin | | Guil basin | |
|---|---|---|---|---|---|---|
|  | $\sigma_P$ | $\sigma_T$ | $\sigma_P$ | $\sigma_T$ | $\sigma_P$ | $\sigma_T$ |
| Free | 0.3 | 0.3 | 0.3 | 0.8 | 0.3 | 0.5 |
| Q assim | 0.4 | 0.4 | 0.5 | 1.1 | 0.4 | 0.5 |
| (Q,FSC) assim | 0.8 | 0.8 | 0.6 | 1.2 | 0.4 | 0.6 |
| (Q,CRS) assim | 0.8 | 0.8 | 0.6 | 1.2 | 0.4 | 0.6 |
| (Q,CRS,FSC) assim | 1.0 | 1.0 | 0.8 | 1.4 | 0.5 | 0.7 |

**Table 1.** AR1 processes parameters applied on precipitation $(\phi_P, \sigma_P)$ and temperature $(\phi_T, \sigma_T)$ forcings, for the free ensemble and the
assimilation ensembles during the reanalysis period (September to March) and the forecast period (April to August).

## 4.3 Assimilation setup

The assimilation is performed using a particle filter (PF) with sequential importance resampling (Gordon et al., 1993; Van Leeuwen,
2009). The PF determines sequentially, within an ensemble of simulations (also called particles or members), the simulations
having a model state close to the observations. The PF describes the prior probability density of the system state as a Dirac sum
of equal weights 1/N for N the size of the ensemble. Using Bayes' theorem, the analysis assigns larger weights to the simula-
tions closer to the observations. The weights are then used to resample the simulations farthest from the observations so that
the simulations closest to the observations are duplicated. In this study, we use a stratified resampling method introduced by
Kitagawa (1996). The duplicated simulations are not perturbed after resampling. The dispersion of the ensemble is maintained




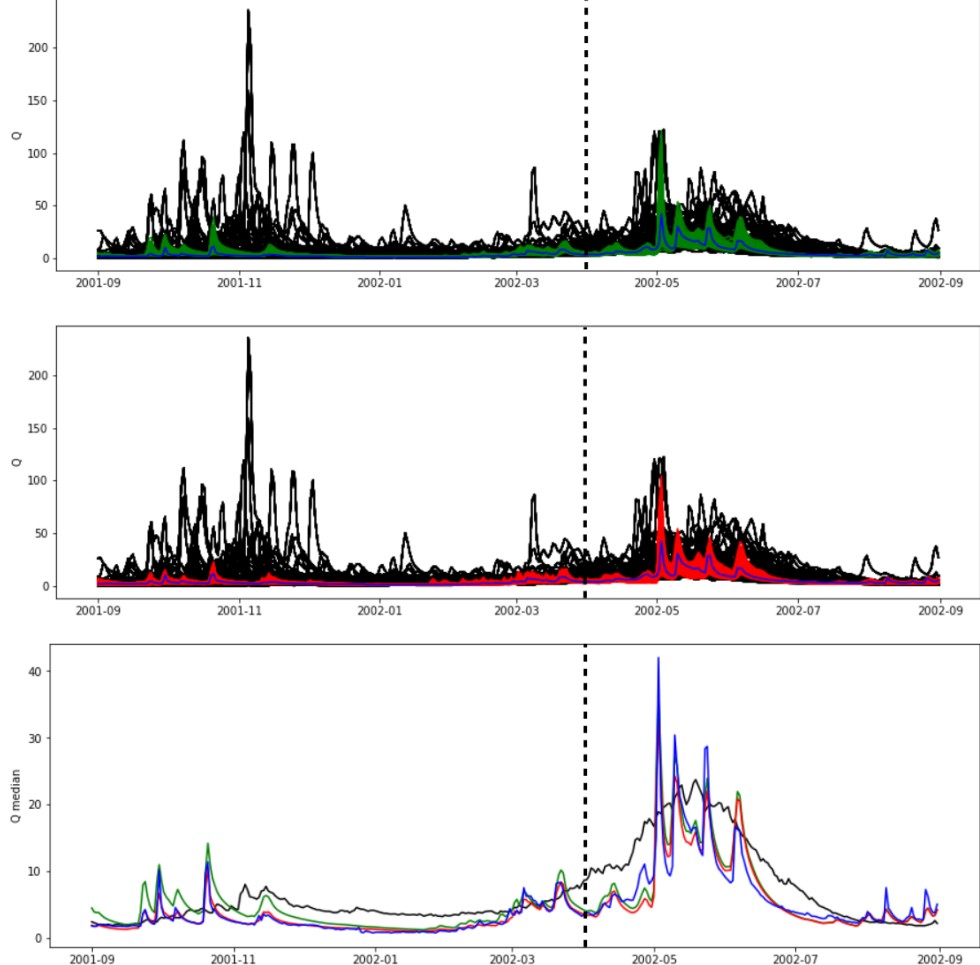

**Figure 9.** Streamflow time series Q, during the year 2002 for the Verdon basin, of the observed streamflow (blue), the climatological ensemble (black), the free ensemble (top panel ; green) and the assimilated ensemble (middle panel ; red) for the assimilation of Q. Bottom panel represent the ensemble respective medians. The vertical dotted black line represents the separation between the reanalysis period (before the line) and the forecast period (after the line).

only by the perturbations on the forcings. Several studies showed the need for additional perturbation after resampling in order to avoid ensemble collapse yet it does not seem necessary in our system.

The assimilation window for all experiments is a 3-day window, i.e., an analysis is performed every three days using the last three daily observations.





|  | Verdon basin | | Naguilhes basin | | Guil basin | |
|---|---|---|---|---|---|---|
|  | Free | Q assimilation | Free | Q assimilation | Free | Q assimilation |
| FinS | 0.701 | 0.665 | −0.015 | 0.218 | 0.755 | 0.830 |
| JustS | 0.371 | 0.796 | 0.311 | 0.440 | −0.366 | 0.410 |
| CRPSS | 0.476 | 0.754 | 0.205 | 0.368 | 0.393 | 0.694 |

**Table 2.** Probabilistic scores on streamflow Q during reanalysis period, from September to March, for the free ensemble (Free) and the streamflow assimilation (Q assimilation).

## 4.4 Observation operators

In order to allow small time lags between simulated and observed streamflow, the three streamflow observations in the 3-day assimilation window are averaged to make a single observation. Streamflow observation error variance is then prescribed as a function of the observed streamflow $Q_{\mathrm{obs}}$ (similarly to Clark et al., 2008; Weerts and El Serafy, 2006; and Piazzi et al., 2021):

$$\sigma_Q^2 = \alpha \cdot Q_{\mathrm{obs}}^2, \qquad (2)$$

with $\alpha = 0.3$. Also, a minimal threshold of $\sigma_Q^2 = 0.2$ is used so as to avoid unreasonably low uncertainties for very small streamflow.

The FSC assimilation is performed using the FSC normalized anomalies. The anomalies are computed by substracting the daily FSC climatologic average to the daily FSC value of the current year and this difference is then divided by the climatologic average. The anomaly indicates with a positive or a negative value if the snow cover is especially high or low this year on that day. The same is done to the fraction snow cover computed by the model. The observation error variances of the FSC normalized anomalies are prescribed at $\sigma_{\mathrm{FSC}} = 0.3$.

Finally, as previously mentioned, CRS observations are local data and do not necessarily represent the snow dynamics of an entire basin. Hence, the first step of the CRS observation operator is to consider the CRS normalized anomalies, similarly to the FSC observations. However, after several tests (not shown here), the CRS normalized anomaly does not provide the correction needed for the model snow stock anomaly at the appropriate altitude band. A second step of the CRS observation operator was then to systematically compare, at each assimilation window, the CRS anomaly to the forecasted model snowpack anomaly at all altitude bands. The closest (in terms of CRPSS) altitude band is then considered to be the observed band. This can be seen as an adaptative observation operator. This process does slightly impact the computation time (as it has to be performed every three days, in this case), but significantly improves the results in our study. The observation error variances of the CRS anomalies are prescribed at $\sigma_{\mathrm{CRS}} = 0.3$.




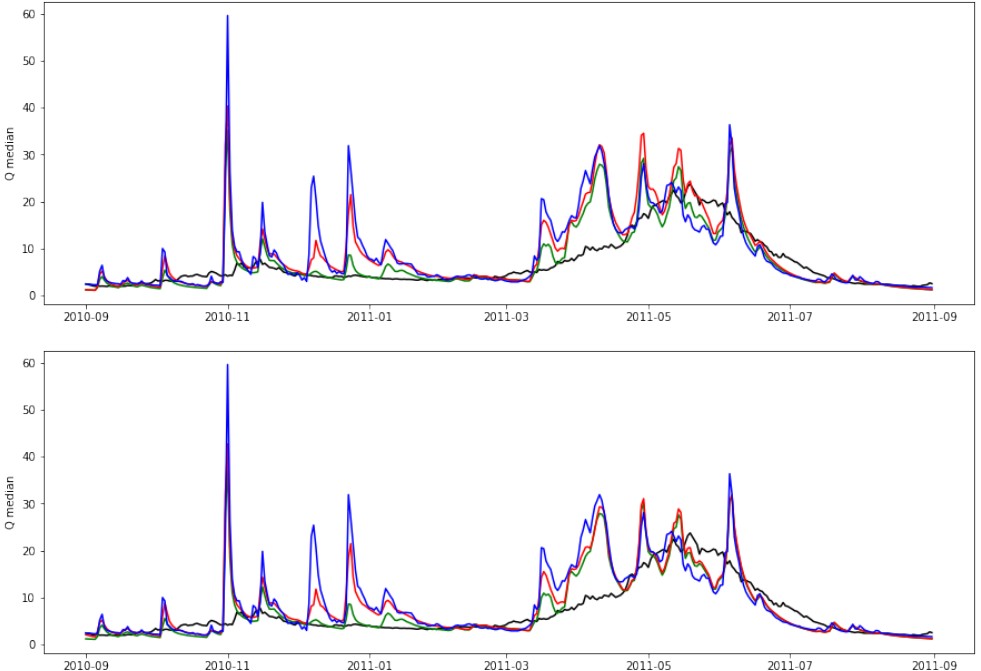

**Figure 10.** Same as the bottom panel of Figure 9, during the year 2011 in the Verdon, for the Q assimilation (top) and the (Q,CRS) assimilation (bottom).

## 5  Assimilation results

### 5.1  Streamflow reanalysis, September to March

During the September to March period, the observations are available daily. In this subsection, only streamflow observations are assimilated. As an illustration, Figure 9 shows the time series of Q during the year 2002 in the Verdon basin. The reanalysis

period corresponds to the times left of the vertical dotted black line and the forecast period to the times right of that line. While the top two panels highlight the high confidence of the assimilated ensemble (red lines) versus the free ensemble (green lines) with a reduction in dispersion, the bottom panel shows that the median after assimilation (red line) is more accurate than the median without assimilation (green line) with respect to the observations (blue line).

The first conclusions drawn from the year 2002 are confirmed over the 16 years 2002-2017 in the Verdon, the 12 available

years between 2004 and 2017 in the Naguilhes and the 10 available years between 2005 and 2016 in the Guil basin, with the use of the probabilistic score CRPSS and its components FinS and JustS summarized in Table 2. The FinS of the free ensemble is higher than the FinS of the assimilated ensemble which is not abnormal since the ensembles have not been generated with the same perturbations and the assimilated ensemble perturbations were much stronger. However, the assimilation increases the JustS of the free ensemble from 37.1% to 79.6% in the Verdon, 31.1% to 44% in the Naguilhes and -36.6% to 41% in the





Guil basin. This results in a CRPSS of 75.4%, 36.8% and 69.4% after assimilation when the free ensemble CRPSS was 47.6%, 20.5% and 39.3% in the three basins, respectively.

Assimilation of streamflow observations combined with CRS and FSC observations have been compared to streamflow-only assimilation and has very little to no impact on the results during this reanalysis period (not shown here). This is due to the very straightforward task of constraining simulated streamflows using the accurately observed streamflow. Indeed, the PF

sequentially selects and resamples the simulations with a streamflow closer to the observations.

An interesting specificity of the particle filter, as a data assimilation method, is that each time not only the accurate streamflows are selected but also all the corresponding state variables. In other word, one can hope that the assimilation will have also selected more accurate snow stocks which will then help produce better streamflow predictions during the following spring and summer seasons.

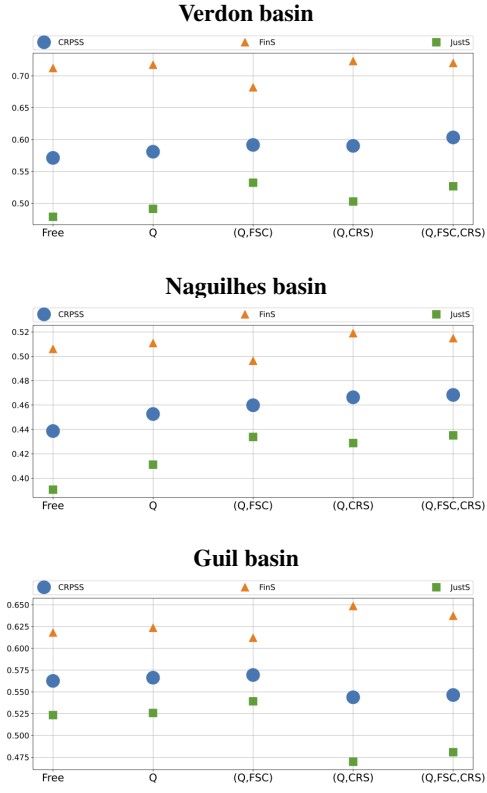

**Figure 11.** Probabilistic scores (bottom) for the forecasted streamflow Q by the free ensemble (Free) and the four assimilation experiments.



## 5.2 Streamflow forecast, April to August

Figure 10 shows the streamflow time series of the ensemble medians (climatologic ensemble in black, free ensemble in green and assimilated ensemble in red) and the observed streamflow (in blue) during the year 2011 in the Verdon basin for the Q assimilation (top panel) and the (Q, CRS) assimilation (bottom panel). The streamflow assimilation (Figure 10 top panel) seem to improve the short term (first 5 to 10 days) streamflow forecast. But, the streamflow forecast is then overestimated after a couple of weeks. However, a good control of the snow pack with (Q, CRS) assimilation (Figure 10 bottom panel) reduces this long term streamflow forecast overestimation. Hence, the overall streamflow forecast remains improved in the first few weeks of the forecast period in comparison to the free ensemble and the overestimation during the rest of the forecast period is avoided.

This result is confirmed by the probabilistic score CRPSS for all years available and in two of the three basins: Verdon and Naguilhes basins (Figure 11). Both (Q,CRS) and (Q, FSC) assimilation show CRPSS increase in comparison to Q only assimilation.

In the Guil basin, the streamflow forecast is degraded by the use of CRS data. This overall result is in fact due to some years in particular where the CRS information is contradictory to the observed streamflow. The case of the Guil basin will be further discussed in the following section.

In general, the improvement brought by an accurately controlled snow pack to the streamflow estimation is slim in terms of scores. The variation in the CRPSS is almost always smaller than 5%. This is due to the fact that, even though the snow stock might be improved, the timing of the streamflow runoff is mainly driven by the anticipated meteorological forcings during the forecast period. The cumulated streamflow, or seasonal streamflow supply, however should be less impacted by the timing of the runoff and should be significantly improved by a better estimated snow stock.

## 5.3 Seasonal streamflow supply A48 forecast

Global scores have been computed to assess the abilities of the different assimilation configurations to estimate the seasonal streamflow supply A48, i.e., the cumulated runoff between April and August. Similarly to Figure 11, Figure 12 shows the global CRPSS, FinS and JustS of the A48 ensemble estimations in the three basins and Figure 13 shows the RMSE of the ensemble means.

As for the streamflow estimation, the A48 estimation is improved by assimilating all the available data in the Verdon basin and the Naguilhes basin. In the Verdon basin, the Q assimilation increases the CRPSS from 77% (free ensemble) to 80.5% and the (Q, FSC, CRS) assimilation further increases the CRPSS to 82.7%. Meanwhile, the RMSE of the free ensemble A48 mean (200 hm$^3$) is almost halved by the (Q, FSC, CRS) assimilation and reduced to a little over 100 hm$^3$. In the Naguilhes basin, the Q assimilation increases the CRPSS from 58.7% (free ensemble) to 62.2%, the (Q, CRS) assimilation further increases the CRPSS to 75% and the (Q, FSC, CRS) assimilation CRPSS is slightly lower at 74.6%. The RMSE is strongly reduced by the use of CRS observations. Both (Q, CRS) and (Q, FSC, CRS) assimilations reduce the free ensemble A48 mean RMSE, which is over 10 hm$^3$, down to under 4 hm$^3$.



The yearly A48 CRPSS histograms presented in Figure 14 and Figure 15 allow to understand how combining all the observations improve the global scores. Every year, the free ensemble A48 CRPSS (green) is compared to the A48 CRPSS of the assimilated ensemble (red) for the different assimilation configurations. As a reminder, a negative CRPSS indicates that the ensemble estimation is less accurate (in terms of CRPS) than the climatological ensemble. In the Verdon basin, the main inaccurate free ensemble A48 estimation occurs in 2014. During that year, only the assimilation configurations containing CRS observations are able to truly correct that estimation. Meanwhile, in 2003 for instance, (Q, CRS) assimilation deteriorates the A48 estimation. But, only assimilating (Q,CRS,FSC) manages to improve both 2003 and 2014. Similarly, in the Naguilhes basin, the years 2004 and 2011 are poorly estimated by the free ensemble, the Q assimilation and the (Q, FSC) assimilation but both the (Q, CRS) and the (Q, FSC, CRS) assimilations significantly increase the A48 CRPSS. These yearly variations show that the hydrological problem considered here is not a linear and Gaussian problem where adding observations systematically improves the estimation every year. These variations can be due to the quality or representativity of those observations which can also variate from one year to the next. But in this case, the benefits of combining multivariate data comes from the particle filter selection process which behaves as a data cross validation of sort that will take advantage of the most appropriate observations each year. This remains true, however, as long as none of the observations are widely inaccurate.

For instance, in the Guil basin, the detrimental impact of the CRS observations is even larger on the A48 estimation than it is on the streamflow estimation. The average CRPSS declines from approximately 75% to 65% (Figure 11) and the RMSE increases from 120 hm$^3$ up to 280 hm$^3$ (Figure 12). Similarly to Figure 14 and Figure 15, the yearly histograms in Figure 16 reveal that the assimilation performances differ significantly from one year to the next. In particular, the histograms reveal that the A48 estimation in 2012 is improved by the CRS observations. However, for the years 2008, 2013 and 2014, the CRS observations seem to completely mislead the A48 estimation. This behavior can be explained by specific atmospheric events during those years rending the CRS observations unrepresentative of the hydrological situation of the basins. In particular, for the year 2008, the poor results can be explained both by the non-representativity of the CRS observations and by a historical flooding episode at the end of May, associated with strong uncertainties on precipitation.

## 6 Summary and conclusions

The objective of this work is to assess the potential of using local snow observations such as cosmic ray sensor observations (CRS) and fractional snow cover (FSC) data in order to improve the estimation of the seasonal streamflow supply at the outlet of mountainous basin. The assimilation of streamflow measurements is commonly performed and is known to improve the short term prediction of hydrological system evolution. However, combining different snow pack observations at basin-scale, such as FSC data, and at local scale, such as CRS data, could lengthen the prediction time, so much so that the prediction of seasonal streamflow supply between April and August, the A48, can become reliable.

As a first step, a sensitivity test performed in Section 3 shows that snow stock control has the potential to strongly reduce the uncertainties on the seasonal streamflow supply. The Sobol indices (relative variances) demonstrate a significant sensitivity of the seasonal streamflow supply A48 to uncertainties on the snow stocks at different altitudes ($S_4$ to $S_8$ in the Verdon, $S_2$



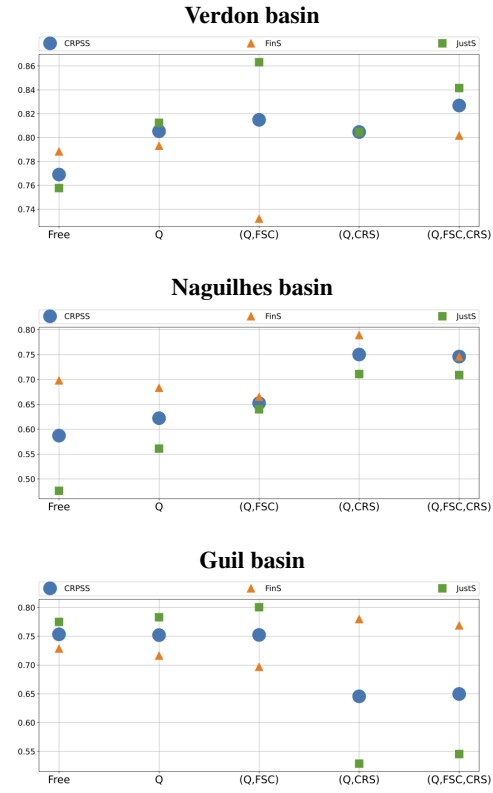

**Figure 12.** Probabilistic scores for the forecasted seasonal streamflow supply A48 by the free ensemble (Free) and the four assimilation experiments.

to $S_4$ in the Naguilhes and $S_4$ to $S_7$ in the Guil basin). Although expected, this result supports the idea that assimilating data containing snow stock information can improve A48 estimation.

The streamflow assimilation is confirmed to be beneficial for the streamflow estimation during the reanalysis period from September to March (Section 5.1), the streamflow prediction (after the assimilation) from April to August (Section 5.2) and
for the A48 estimation (Section 5.3). Indeed, in the three basins, the Q assimilation significantly improves the streamflow estimation during the reanalysis period: from 47.6% to 75.4% in the Verdon basin, from 20.5% to 36.8% in the Naguilhes basin and from 39.3% to 69.4% in the Guil basin. Also, the streamflow forecast during the 5 month forecast period is systematically improved by the Q assimilation, albeit slightly. Finally, the A48 CRPSS of the Q assimilation is increased compared to the one of the free ensemble for both the Verdon and the Naguilhes basin and remains approximately constant in the Guil basin. In all
the basins, the A48 RMSE of the Q assimilation ensemble mean is smaller than the free ensemble mean.

Moreover, in two of the three basins, the addition of FSC and CRS in the assimilation process has proven to be very beneficial to the A48 estimation. In the Verdon and the Naguilhes basins, where it was identified that FSC and CRS are beneficial, the best strategy seems to be that all available observations are to be included in the assimilation process so that if some years





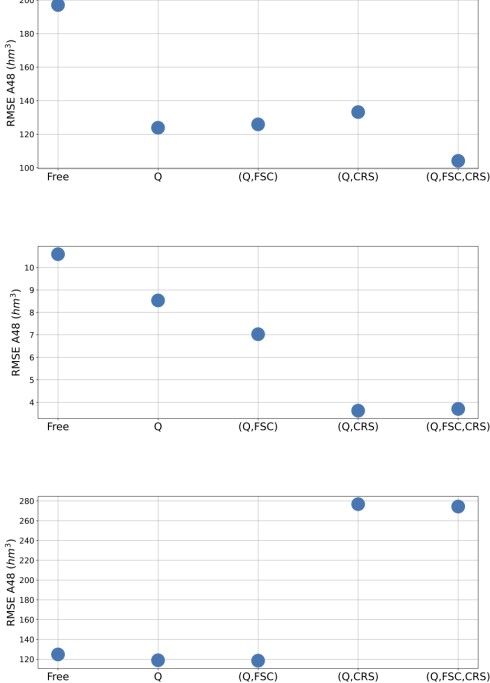

**Figure 13.** RMSE for the forecasted seasonal streamflow supply A48 by the free ensemble (Free) and the four assimilation experiments. Not shown here, the climatology RMSE are: 3603.43 hm$^3$ in the Verdon basin, 57.26 hm$^3$ in the Naguilhes basin and 2079.76 hm$^3$ in the Guil basin.

one observation type is misrepresenting the hydrological situation the others can counteract its effect. For instance, in the

Verdon basin, the (Q, FSC) assimilation degrades the estimation in 2014 and in 2003 the (Q, CRS) assimilation degrades the estimation but when assimilating (Q,CRS,FSC) both those years are improved. The overall scores show that the (Q, FSC, CRS) assimilation when compared to the Q only assimilation increases the A48 estimation CRPSS by 2% in the Verdon and by 22% in the Naguilhes basin and reduces the A48 estimation RMSE by 19.7 hm$^3$ in the Verdon, corresponding to a 15.9 % improvement, and by 4.8 hm$^3$ in the Naguilhes basin , corresponding to a 56.6 % improvement.

A caution must be made on this strategy since, in the Guil basin during specific years, the CRS observations can be largely misrepresenting the hydrological situation thus significantly deteriorating the streamflow and A48 estimations. More specifically, during three years (2008, 2013 and 2014) the CRS observations appear to be in contradiction with the streamflow observations hence misleading the A48 estimation. This can be explained by specific atmospheric events occurring during those years and leading to highly heterogeneous precipitation patterns.

The present study showed that assimilating multivariate data in a basin scale hydrological model is possible and can improve long term predictions such as the seasonal streamflow supply estimation. In two of the three basins, the assimilation of snow




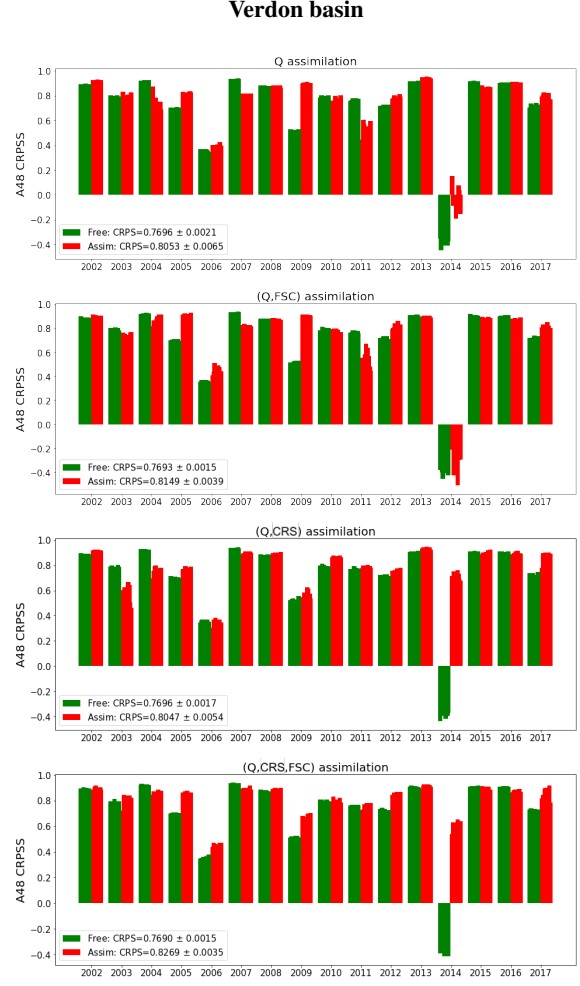

**Figure 14.** Yearly CRPSS of the seasonal streamflow supply A48 for the free ensemble (green) and the assimilated ensemble (red) from the Q assimilation, (Q,FSC) assimilation, (Q,CRS) assimilation and (Q,CRS,FSC) assimilation (10 experiments each year) in the Verdon basin.

observations has proved beneficial, improving the overall performances. This result was achieved by incorporating local CRS data into a basin model through the use of an adaptive observation operator on the elevation band. Albeit heuristic, the adaptive observation operator has proven to be successful in most cases. However, some years, the poor representativity of local CRS

observations can degrade the performance of the DA process. Combining the sources of observations therefore appears to be the best guarantee of robustness for operational purposes. Also, the multivariate assimilation allowed to highlight that the CRS observations in one of the studied basins and during specific years are not appropriate for assimilation and should be disregarded.



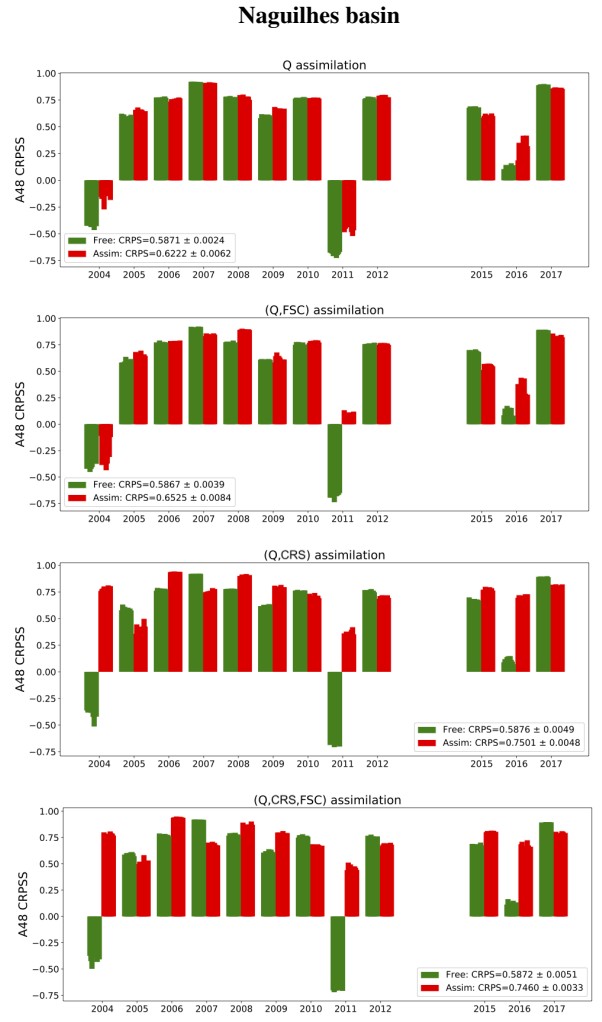

**Figure 15.** Same as Figure 14 in the Naguilhes basin.

As a continuation of this work and to keep improving A48 prediction in operational forecasting systems, several other aspects must be further investigated. First, a wider study should be conducted using the same experimental set up to assess the benefits and issues of the available observations, in particular the CRS data, in a larger panel of hydrological basins. Options to compensate for the lack of representativity of the CRS data in some basins are limited. A study on the cost-benefit to densify the observation network in the concerned basins should be conducted by operational centers. A more attractive, because less expensive, alternative could be to better characterized and/or improve the representativity of SWE data at basin scale by using the existing large network of snow poles that may contain complementing SWE information. Finally, moving from the semi-







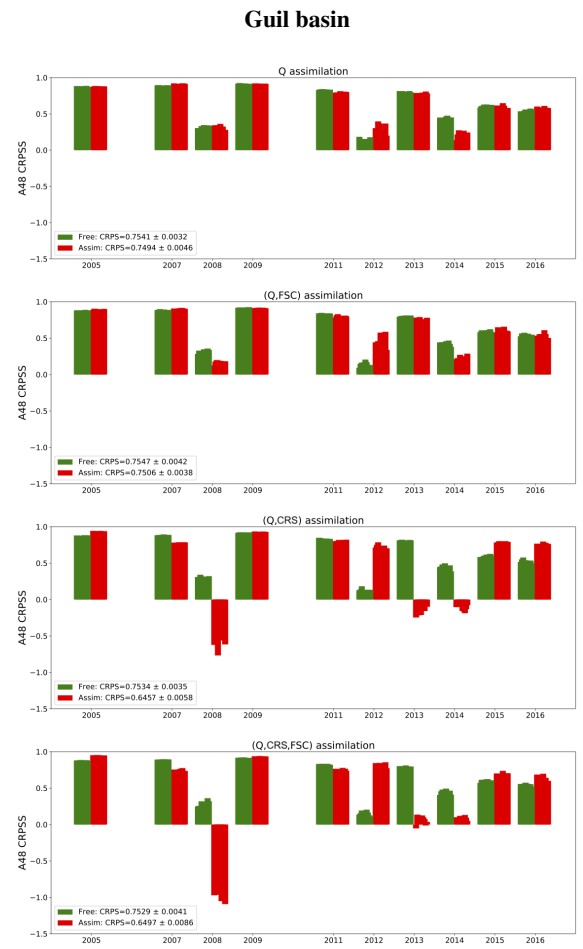

**Figure 16.** Same as Figure 14 in the Guil basin.

distributed MORDOR model to the fully spatialized MORDOR model (Rouhier et al., 2017) should make the integration of local CRS information into the physics of a basin model more realistic and ultimately improve the A48 estimation.

*Data availability.* The datasets generated for this study are available on request to the corresponding author. The codes are of commercial use and can not be shared.

*Author contributions.* All authors designed the study. Sammy Metref and Emmanuel Cosme designed the numerical experiments. All authors contributed to the analysis of the results. Sammy Metref led the redaction of the manuscript.



*Competing interests.* The authors declare that the research was conducted in the absence of any commercial or financial relationships that could be construed as a potential conflict of interest.

*Acknowledgements.* This research was funded by EDF - 44200965 - DMM (project number H-44200965-2017-000363-A).



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
