# Peer review of "Snow data assimilation for seasonal streamflow supply prediction in mountainous basins"

_EGUsphere, 2022_

## Author Response (AR1)

**Editor's decision**

**Dear authors,**

**The reviewers find the paper interesting, but major changes are needed. I concur with this assessment. Please address all review comments in the revised paper along the lines of your response. However, it appeared to me that some of the comments of reviewer 2 have not been fully addressed, so please make an attempt to do so.**

*We would like to thank the Editor for overseeing the revision process.*

*We are not sure, however, which of the comments of Reviewer 2 the Editor is referring to. We have tried to provide more extensive answers to several points and hopefully the combination of the "response to the reviewers" document with the updated manuscript will shed more light on our answers. We would be happy to provide further details and/or clarifications to specific comments if need be.*

**Response to the reviewers**

**Reviewer 1**

We would like to thank Reviewer 1 for the constructive comments. Please find below a point-by-point response to the review. Most of the minor comments have been addressed directly in the manuscript (highlighted in red). In the following, we address (in italic font) the major and moderate comments of the Reviewer (in bold font).

**The paper aimed at using real data assimilation to assess how much the streamflow supply prediction can be improved by assimilating additional snowpack information. The performance of directly assimilating streamflow (Q), fractional snow cover (FSC) and cosmic ray snow sensor data (CRS) and their combination are assessed in three basins with first Sobol sensitivity analysis and then real streamflow data. The authors found that Q assimilation notably improves streamflow estimations during both reanalysis and forecast period, while additional combination of CRS and FSC data to the assimilation further ameliorates the A48 prediction in two of the three basins. Overall, the topic is of interest to operational streamflow forecasting for snow-dominated areas. The data and methods are reliable, and the results are supporting what's concluded. However, I have some major frustrations with the writing, the main figures, and some moderate concerns with the methodology, as shown below. This should warrant at least major revision.**

**Major:**

1. **The Introduction will benefit from a re-structuring. For example, Line 28-30 and Line 63-65 present research questions and objectives of this study, which can be combined. Also, the authors seem to have mixed their results with the Introduction (see Line 35 and Line 75 for examples of unnecessary results), which are usually presented in the conclusion or discussion part. I suggest authors to overhaul their Introduction to give clearer outlines.**

*The introduction has been restructured. In particular, the results paragraph has been removed, the research questions and objectives have been gathered in a single paragraph and a paragraph on the role of uncertainties in our study has been added. The introduction is indeed clearer and more impactful now.*

2. **I think there are major issues with many figures. Fig. 1: Resolution is too low, and subscripts are not recognizable. Using figures from past studies is okay, but needs better caption to describe each individual term appearing in the figure (e.g., what is AEP, what is PET? Seems Potential ET is used as a forcing but not described in your model description part).**

*Figure 1 has been removed because it was not essential. We have decided to provide a more detailed model description in Section 2.1 and to refer the reader to the Garavaglia et al. (2017) paper for further details. The Potential ET (EvapoTranspiration) can be expressed as a forcing but is often parametrized as a function of temperature instead of an independent external forcing. Garavaglia et al. (2017) describe this parametrization.*

    **Fig. 2 lacks description about the shading. Please add a legend.**

*A legend has been added to Figure 2 (now Figure 1). The shading is in fact multiple gray curves, for the 50 ensemble members, on top of each other.*

    **Figs. 5-7 lack the description about the variables plotted (there are need to describe them both in the caption and the main texts). This type of figure presentation is difficult to be accepted by the academic community. Suggest authors to re-draw many figures.**

*A description of Figure 5-7 has been added to both the caption and the main text. The caption now describes the figure and provides a better understanding of the results.*

3. **Lack of variable descriptions in Figs. 5-7 is preventing readers from clearly getting your methodology: which ones are the most sensitive? (see comments above)**

*Same answer as previously. The caption and the main text now contain indications on how to understand and interpret the results of this sensitivity experiment. The uncertainty on the snow stocks at level 4 to 7, in the Verdon and the Guil, and at levels 2 to 4, in the Naghuiles, generate the highest Sobol indices. Meaning, they are the variables that generate the most uncertainty on the seasonal streamflow supply.*

4. **I think in the Method section, it is lacking the spatial plots for the FSC and CRS measurements locations. These are key information (how much? Where are they located?)**

*Figure 3 (now Figure 2) has been replaced with a new figure that includes a zoom on each basin and displays the location of in-situ CRS observations.*

[Figure]

5. **About Methods: the DA framework/perturbation and the model are relatively better presented. But how about the measurements? How are the FSC and CRS obtained? What about their uncertainty? How about their available number and spatial distribute (this is asked above)? I see some information is presented in Intro, but measurements uncertainty is the most important, and should receive a much more balanced writing and description in a specific Method section.**

*We acknowledge the lack of information on observations and the uncertainty related to their location and measurements. Although, the stance taken in this paper is to regard the observations as they are made available to operational centers, some more information on observation uncertainties and how they are taken into account in the assimilation process has been added to Section 2.2. However, the main source of uncertainty is most likely due to the spatial representativity of the observations with regards to the semi-distributed model representation than from measurement errors.*

6. **Line 181: not sure how is the 900-member ensemble determined? Usually, we use much less ensemble members than this in DA studies. I understand**

**the computation demand may be low for your hydrologic model, but scientifically why is this large number needed? Any justification and supporting evidence on how this satisfies your research goal? If there's a need for inflating the uncertainty, this should be clearly clarified. It may be tested results to maximize performance in Figs. 9-10, but I think understanding the uncertainties (as denoted by the spread of your ensemble) is more crucial to DA rather than to maximize performance.**

*We use a large ensemble size to ensure experimental reproducibility, which was not ensured with smaller ensemble sizes due to high nonlinearities and thresholding steps in the model. The focus of the paper is on the information content of snow observations, not the assimilation performance, so we did not consider the ensemble size issue as a point to investigate in detail.*
*As this point is very pertinent for many users and was not clear in the first version of the manuscript, a small paragraph has been added to discuss it (lines 238-244).*

**Moderate concerns:**

1. **For the Sobol indices equation, it would be better if the authors can provide detailed explanations of the variables in the cases of temperature and precipitation forcing. Also, the equation takes the presumption that the variables are independent and has known probability distributions. However, in geographical analysis it is often hard to determine whether a variable is completely independent. It would be more convincing if the authors can provide some assumptions and preconditions.**

*The Sobol equations indeed make the assumption of independence. However, the Sobol indices are not used here as an attribution diagnostic, where we would need to disentangle the variable dependencies to attribute the contribution of each variable, but as a controllability diagnostic. In this sense, the Sobol indices can help us conclude that if we manage to reduce the uncertainty on the snow stocks (with the use of observations) we will be able to reduce the uncertainty on the seasonal streamflow supply regardless of the original cause of these uncertainties. A paragraph has been added to the manuscript to clarify the Sobol equations' assumptions and to explain the intent with which we use these equations (lines 173-177).*

*Also, a more detailed explanation of the variables behavior in response to the forcings has been added in the manuscript (lines 181-188).*

2. **What is the resolution of the reanalysis data used in the paper? When assimilating observation data, will the resolution differences cause uncertainty and what is the solution used by the authors?**

*The model used here is semi-distributed hence only one temperature and one precipitation is prescribed on the overall region. This simplification is the source of strong uncertainties. The temporal resolution of the forcings is daily (l. 102-104). For MORDOR-SD model, the required input data are a representative estimate of areal precipitation and air temperature. Two orographic gradients are then used for describing the meteorological spatial variability at the elevation zones scale. The precipitation and temperature reanalysis are available at 1-km/1-day resolution, and are therefore averaged at the catchment's scale. The impact of the forcings resolution has not been investigated but it is likely that the main source of uncertainty on the meteorological forcings remain spatial due to the semi-distributed nature of the model. As stated in the paper, observations are in-situ and assimilated in a semi-distributed model, which leads to inevitable representativity errors.*

**Minor ones:**

1. **Line 6. 'Lead to' mis-spelled as 'leed to'.**

*This mistake has now been corrected in the article.*

2. **Line 10. 'A series of' not 'a serie of'**

*This mistake has now been corrected in the article.*

3. **Line 35: 'play a role in' not 'on'**

*This mistake has now been corrected in the article.*

4. **The subtitles like "Hydrological system" do not exactly match the content.**

*The subtitle is now "Model and observations".*

5. **Line 311. Is the "prediction time" the same as the forecast period mentioned in line 72? How could the prediction time be lengthened?**

*The sentence at line 311 was poorly formulated. The sentence should not have said that the prediction time is lengthened but rather that the seasonal streamflow supply prediction can be improved. A new formulation is now proposed at lines 359-360.*

6. **9 and 10: to improve readability for the readers, please add the legend directly to the plots.**

*Legends have now been added to the plots.*

7. **What exactly does A48 represent? In line 2 it seems to men "between April and August", while in line 20 it seems to stand for seasonal streamflow supply. This type of writing will confuse readers.**

*The paper now defines the acronym SSS as the seasonal streamflow supply that is computed as the accumulated streamflow between April and August. Hopefully, the confusion is now lifted.*

**Reviewer 2**

We would like to thank Reviewer 2 for the constructive comments. Please find below a point-by-point response to the review. Most of the minor comments have been addressed directly in the manuscript (highlighted in red). In the following, we address (in italic font) the major and moderate comments of the Reviewer (in bold font).

**The manuscript 'Snow data assimilation for seasonal streamflow supply prediction in mountainous basins' by Metref et al. provides an interesting study regarding the big challenge of improving streamflow predictions in mountainous, snow-dominated regions. The authors investigate the additional value of directly assimilating streamflow, fractional snow cover (FSC) and SWE measurements (taken by cosmic ray sensors (CS)) and their combinations in terms of improving seasonal forecast in three French basins (one in the Pyrenees and two in the Alps) applying a conceptual semi-distributed hydrological model (MORDOR-SD) as basis. They test their results during reanalysis (assimilation) and forecast periods and found that (not surprisingly) the assimilation with streamflow improved the estimates during both, reanalysis and prediction. Including CRS and FSC to the assimilation process could further improve the seasonal prediction in two of the three catchments.**

**In general, this topic is interesting to the readers of the journal. However, the manuscript in its current state needs major improvements before considering for publication. The authors should add important additional information and clarifications at several points (see comments below) as the paper currently produces several question marks in the eyes of the reader at some points. I agree with the points raised by reviewer 1. In addition, I have further points, which are listed below. English language should be improved.**

**General comments:**

- **In your study, you are focusing on snow-dominated catchments where the simulation of snow processes plays an important role. However, the description of the snow module of the MORDOR-SD**

**model is entirely missing here (e.g., I guess it is a simple day-degree approach to describe snow melt). This should at least be described (Section 2) and discussed (Section 5 or 6) carefully.**

*The snow model is indeed derived from a classical degree-day scheme, with a few important additional processes: (i) a cold content able to dynamically control the melting phase; (ii) a liquid water content in the snowpack; (iii) a ground-melt component; and (iv) a variable melting coefficient, depending on the potential radiation assumed to model the changing albedo effect throughout the melting season. The accumulation phase is controlled by the discrimination of the liquid and solid fractions of the precipitations. A description of the snow module has been added in Section 2.1.*

- **In general, assimilation can lead to good results regarding streamflow predictions (as you have shown). However, it should also be discussed in the paper, if adding more physical realism in describing snow cover processes could also lead to improved results regarding streamflow predictions.**

*Adding more physical realism to the snow cover processes would perhaps improve the streamflow predictions. For the reviewer's information, in the context of operational prediction, EDF teams do work on improving streamflow predictions by simultaneously upgrading the physical realism of their models and enhancing their assimilation capabilities. However, this question is not in the scope of the paper and, since the experiments we perform do not provide information on that topic, adding a discussion would be mainly speculation on our part.*

- **As reviewer 1 already stated, the introduction is difficult to read and a mix of state of the art, presentation of some results, objectives, research questions, outline, and some methods. The Introduction should be carefully improved including a solid state of the art paragraph.**

*The introduction has indeed been restructured. The results paragraph has been removed, the research questions and objectives have been gathered in a single paragraph and a paragraph on the role of uncertainties in our study has been added.*
*Thanks to both Reviewers' comments, the introduction is now clearer and more impactful.*

- **What is the reason for selecting the three chosen basins Verdon, Naguilhes, and Gui? Are they very different in terms of topography, meteorology, geology, etc. to learn different behaviours regarding catchments response? Do you expect to gain additional information, if you would select further catchments out of the 50 catchments operated by EDF?**

*These three catchments were selected according to two criteria: (i) the quality of the hydrometric data (to avoid assimilating poor quality data); (ii) the presence of CRS data*

*on the basin. Moreover, they offer an interesting variety of hydro-climatic contexts. This explanation has been added to the article (l. 120-122).*
*Indeed, the Naguilhes basin is small at relatively low altitudes (max. 2750m) which most likely explains the very direct link between snow content and SSS estimation (link highlighted by the Sobol experiment). The Verdon and Guil basins are very similar in surface area (404 and 418 km2) and elevation, however, the Verdon is a very narrow but long basin while the Guil is wide. This difference could be a reason for the poor observation representativity of the CRS measurement in the Guil basin, as shown by the poor assimilation results it generates.*

*This selection would obviously deserve to be extended (as suggested in the discussion), but it already allows us to clearly identify the potential and limits of our assimilation strategies.*

- **You tried out the settings of assimilating i) Q, ii) Q and FSC, iii) Q and CR, and iv) Q, FSC and CR. Why didn't you show the results of just assimilating FSC (regarding CR you stated it would deteriorating the system estimation (l.42ff – this however, would fit rather in a discussion section instead of the intro))?**

*As previously stated, the introduction has been modified and references to these results are no longer made in the introduction. The FSC-only assimilation provides very poor results every year and for every diagnostics. This result is not surprising, as this variable is only indirectly correlated to the snow water equivalent in the basin. Moreover, the goal of the present study is to show the improvement that direct and indirect snow observations can bring to the existing operational streamflow assimilation. We have decided not to show these results for the sake of clarity and to lighten the article. This behavior is now noted in the Summary and Conclusions section.*

- **How well does the MORDOR-SD model perform in general in your catchments regarding calibration and validation periods (e.g., according to objective functions such as Nash Sutcliffe Efficiency)?**

*Modeling performances are pretty good in these basins. For example, hereafter the values of Kling-Gupta and Nash-Sutcliffe Efficiencies on available periods (calibration periods are respectively: 1998-2013 for Verdon, 1987-2012 for Naguilhes, 2004-2013 for Guil).*

|  | KGE | NSE |
|---|---|---|
| Verdon (1997-2017) | 0.921 | 0.846 |
| Naguilhes (1987-2017) | 0.880 | 0.760 |
| Guil (2004-2017) | 0.961 | 0.926 |

*Observed (blue) and Simulated (red) long-term mean daily streamflows are also illustrated:*

*Guil*

[Figure]

*Verdon*

[Figure]

*Naguilhes*

[Figure]

- **I miss the link of meteorological and snow conditions at certain years in the three catchments and your results (especially in the discussions in Sections 5 and 6). Snow and meteorology conditions can be quite different throughout the years and might affect the quality of your streamflow predictions. What is the impact on e.g. a lot of snow vs. shallow snowpack during single winters in the three catchments?**

*The performance of the forecasting system is indeed very variable from year to year. Some meteorological situations (very heterogeneous or localized episodes) can lead to significant precipitation estimation errors and degrade the forecasts. For the three catchments considered, no clear connection between the amount of snow accumulated during the past winter and the SSS prediction has been detected.*

- **In the lines 28-30 you raise three questions on the relevance of using in satellite and situ snow observations to improve seasonal streamflow predictions in mountainous catchments. However, I have the feeling that these questions are not properly answered in the course of the manuscript.**

*We agree that the third question (How do existing analysis methods perform in estimating the snowpack from the snow observations?), as formulated, may appear to go beyond this study. We decided to remove it from the introduction.*

*Considering the first two questions:*

*(i) How much is the SSS prediction sensitive to the snowpack? We attempt to answer this question using the Sobol analysis presented in section 3, which demonstrates that SSS prediction is primarily sensitive to snowpack water content.*

*(ii) Do the snow observations contain enough information to estimate the snowpack accurately to impact the quality of predictions? Results of the study demonstrate that assimilating snow observations (FSC and SWE) in combination with streamflow observations improve SSS prediction, compared to assimilation of streamflow only.*

**Specific comments (chronologically):**

- **86: The paper is actually divided into six parts (including the introduction). -> Better reformulate: 'The paper is structured as follows:'**

  *Corrected in the text.*

- **Figure 1 and Section 2.1: At least a basic introduction to the model and its components should be given. Please add information (e.g., as a**

**legend in Figure 1) on the variables and parameters shown in the graph.**

*As discussed in Reviewer 1 comments, Figure 1 has been removed. A model description has been added in Section 2.1 and the reference to Garavaglia et al. (2017) is provided for a detailed model description.*

- **93: How many metres does one elevation band encompass? Regarding Figures 5-7, this seems to be 250 m for each catchment!?**

*Classically, the number of elevation zones is optimized depending on the hypsometric curve of the catchment according to the following criteria: (i) the relative area of each elevation zone has to be greater than or equal to 5% and less than or equal to 50 %, and (ii) the elevation range of each zone has to be lower than 350 m. It leads to 8 elevation bands for Verdon and Guil and 4 for Naguilhes.*

- **95: What are the orographic gradients (lapse rates) applied in this study for temperature and precipitation?**

*The orographic gradients gpz and gtz for precipitation and temperature respectively are now defined in the text (lines 94-95 and 105-107).*

- **99: What are the 5 state variables (I guess S, G, U, L, Z and N?) and the 2 global variables? Please add this information at least in the the methods descriptions and the legend of Figure 1. In additions, why do you write the span of 10-12 free parameters? How many did you have in your setup?**

*The mention of five variables in each elevation band was a reference to the 4 storage water levels U,S,L and Z and the snowpack bulk temperature TST. There is only 1 global variable N representing the deep storage water level. The number of free parameters refers to the model calibration process, ranging from 10 to 12 depending on the site-specific calibration strategy. This is now made clearer in the article (l. 96-100).*

- **106-111: Not entirely clear; Please improve the descriptions in these lines. / Figure 2: Is this a catchment averaged meteorological data set shown here or is it representative for one elevation band? In general, not sure if Figure 2 is really needed. In addition, the Verdon basin is actually introduced one Section later and the reader might be wondering why you already mention it here.**

*This paragraph has been reformulated (line 106-113) and should now be clearer. / Figure 2 shows the catchment averaged meteorological data set which is the only input of the MORDOR-SD model. Introducing stochastic perturbations is crucial for the following experiments, we have hence decided to use an illustration (Figure 2) even though the Verdon basin is not yet described.*

- **Figure 3: This graph just shows the location of the basins in France. The graph misses topographic information as well as important information such as at least the location of its capital and the name of the mountain ranges (Pyrenees, Alps). In addition, I suggest giving a more detailed overview on the three selected basins in the Figure.**

*Figure 3 (now Figure 2) has been updated.  Topographical and geographical information has not been added to the figure to keep it clear, but is given in the text (l. 123-128).*

- **140f: Please add information on the expected footprint of the CRS as well as limitations of this sensor type.**

*The expected CRS footprint is classically about 5m, and although this measurement technique is known to provide accurate SWE estimations (except for very shallow snow-depth), the CRS provides very local information which can be a limitation for basin-scale assimilation. This was added in the text (l. 142-144).*

- **142ff: Please describe in more detail how FSC was derived. Did you look at basin-averaged values or did you consider elevation band based FSC values. I think just taking FSC values for the entire catchments (with elevation ranges of approx. 2000 m) is not sufficient and might not be representative for the application of assimilation data.**

*FSC data used in this study are basin-averaged values. We agree that elevation band-averaged values are potentially more relevant, but they have not been used because they are too incomplete (limiting cloud cover over small areas).*

- **Figures 5-7: Not introducing U, L, Z, S, TST before makes the figures questionable (see comments above). Information regarding elevation bands is missing in the y-axis. The chosen (linear) colour representation is not very meaningful. In addition, I would suggest to add a row in showing the average Sobol indices for the entire time period to get a clearer overall picture. Interpretation why the Sobol indices are higher for some elevation bands as well as distinct years is missing in the text.**

*The state variables U, L, Z, S and TST are now defined and described in the model description and in the Sobol experiment interpretation. Elevation band numbers have been added to the Figures. The colorbar for representing the Sobol indices has been changed, also, the percentages are no longer displayed  from 0 to 100 but from 0 to the maximum Sobol index for each basin. This allows to highlight the overwhelming sensitivity of the SSS to the snow stocks. We have also added a column showing the averaged Sobol indices over the entire time period.*

*The differences between elevation bands is mainly due to the differences of their absolute snow content. For example, high elevation bands have smaller areas (by*

*definition of the elevation bands) hence they have less snow content which leads to less uncertainty.*

*Similarly, differences between years are most likely due to differences in snowfall since the perturbations are prescribed relative to the state variables (in percent) but the sensitivity of the streamflow is absolute.*

- **170-174: Please avoid repetitions – was already introduced before.**

*We consider this experimental setup information important. We decided to mention it in the introduction and give it in more detail in the Protocol and Diagnostics section.*

- **Figures 9 and 10: Please insert for a better readability legends. Why do you show the selected years, assimilation configurations (assim. of Q in Fig. 9, assim. of Q and Q&CRS), and the selected catchment as an example in those Figures as examples? Are other seasons/years similar in their quality?**

*Legends have been added to Figures 9 and 10 (now Figure 8 and 9).*

*These figures are only illustrations. Figure 9 (now 8) illustrates ensembles of simulation with and without streamflow assimilation which relates to Section 5.1. And Figure 10 (now 9) illustrates the improvement brought by the assimilation of (Q, CRS) in comparison to Q assimilation only.*
*The exhaustive study, over the different years and different basins, is provided by the following diagnostics (Figures 10 to 15).*